# Organic Dyes versus Adsorption Processing

**DOI:** 10.3390/molecules26185440

**Published:** 2021-09-07

**Authors:** Francisco J. Alguacil, Félix A. López

**Affiliations:** National Center for Metallurgical Researcher (CENIM), Spanish National Research Council (CSIC), Avda. Gregorio del Amo 8, 28040 Madrid, Spain; fjalgua@cenim.csic.es

**Keywords:** organic dyes, adsorption, removal, wastewaters, environment

## Abstract

Even in the first quarter of the XXI century, the presence of organic dyes in wastewaters was a normal occurrence in a series of countries. As these compounds are toxic, their removal from these waters is a necessity. Among the separation technologies, adsorption processing appeared as one of the most widely used to reach this goal. The present work reviewed the most recent approaches (first half of the 2021 year) regarding the use of a variety of adsorbents in the removal of a variety of organic dyes of different natures.

## 1. Introduction

Several industries need to use organic dyes in their respective processes. However, the usefulness of these dyes, including either natural or synthetic, presents an important odd point: they are toxic for life, thus their presence in waters, wastewaters, and in general liquid effluents must be avoided to obey the regulatory discharge laws for these compounds and to decrease the risk of possible contact with humans.

Various technologies seem to be appropriate for the treatment of organic dye-bearing waters but among them, adsorption processing appears as one of the most used, probably due to its apparent operational simplicity and myriad of potential adsorbents which can be used in this environmental issue.

The importance of these adsorptive methodologies in the removal of organic dyes from waters is reflected in the number of investigations and reviews published in the years. Recently, several reviews about the use of a given type of adsorbents in this role have been published, including silica-based mesoporous materials of the M41S and SBA-n families [1], different types of graphene-based materials [2,3], biochars [4], metal–organic frameworks (MOF) [5,6], various types of polymeric nanofibers [7], metal-doped porous carbon materials [8], and activated carbon fibers [9].

The present work reviews very recent publications (first half of the 2021 year) regarding the use of adsorbents in the removal of organic dyes from waters and, in many cases, the investigations are carried out on synthetic solutions, but there are also some examples in which adsorbents are used on real waters. Readers must be aware that this review does not include publications regarding desorption-degradation processing of these toxic chemicals.

## 2. Adsorbents for the Removal of Anionic Dyes from Waters

A green route was used to prepare ZnO:NiO nanocomposites (Z:N) using the Neem leaf extract as a stabilizing agent [10]. Four different samples with three different ZnO:NiO ratios were prepared, namely 3Z:1N, 1Z:1N, 1Z:3N, and 1Z:1N without leaf extract. The different nanocomposites were used for the adsorption of methyl orange from aqueous solutions and some of the results are summarized in Table 1.

With the adsorbent containing an excess of NiO (1Z:3N), a maximum in methyl orange capacity was reached. It can also be observed that the addition of the Neem leaf extract improved the dye uptake. In all the cases, the uptakes derived by the model (Equation (1)) were in accordance with the corresponding experimental values. The adsorption followed the pseudo-first kinetic equation, which in the published manuscript was written as:(1)$$\log\left({\left[\mathrm{MO}\right]}_{\mathrm{ad},e}-{\left[\mathrm{MO}\right]}_{\mathrm{ad},t}\right)=\log{\left[\mathrm{MO}\right]}_{\mathrm{ad},e}-K_{p1}t$$
where K_p1_ represents the model constant. The procedure followed to reuse the adsorbent was as follows: 0.1 M NaOH (desorption), 0.1 M HNO_3_ (neutralizing), washing with water until pH 7, and lastly drying.

A surface modification of mesoporous carbon (MC), based on a Diels-Alder [4 + 2] cycloaddition, and the multicomponent Radziszewski reactions generated a MC@PIL composite [11]. This adsorbent was used to adsorb Congo red at various experimental conditions including: contact time, concentration, pH, and temperature. The investigation demonstrated that the dye was adsorbed on the composite within 10 min of contact, whereas the adsorption capacity was near 331 mg/g, which was greater than that found with the pristine mesoporous carbon. In this work, the experimental data were fitted to the non-linear forms of the pseudo-second order kinetic model (Equation (2)) and the Langmuir isotherm (Equation (6)):(2)$${\left[\mathrm{CR}\right]}_{\mathrm{ad},t}=\frac{K_{p2}{\left[\mathrm{CR}\right]}_{\mathrm{ad},e}^{2}t}{1+K_{p2}{\left[\mathrm{CR}\right]}_{\mathrm{ad},e}t}$$
(3)$${\left[\mathrm{CR}\right]}_{\mathrm{ad},e}={\left[\mathrm{CR}\right]}_{\mathrm{ad},m}\frac{K_{L}{\left[\mathrm{CR}\right]}_{\mathrm{aq},e}}{1+K_{L}{\left[\mathrm{CR}\right]}_{\mathrm{aq},e}}$$
where [CR] represents the Congo red concentration. Desorption was carried out as written in the published article: “…contained a certain proportion of water/ethanol (45%) and a few drops of 1 M NaOH solution during one hour”. Under continuous use and after three cycles, the desorption rate decreased from 87% to 81% and thus the adsorption capacity decreased from 331 mg/g until near 200 mg/g.

Being a natural, low-cost, and eco-friendly material, naturally occurring alumina particles were used as an adsorbent for the removal of eriochrome black T (anionic dye) from an aqueous environment [12]. The adsorption process can be controlled for by electrostatic attractions, the experimental data were well-described by the pseudo-second order kinetic model, and the adsorption isotherms fitted well with the Langmuir isotherm. The maximum adsorption capacity was found to be 45 mg/g, which corresponded to 71% of the dye removal. The adsorption was endothermic and spontaneous. The desorption procedure and reconditioning of the adsorbent for its reuse followed several steps: (i) desorption with 0.1 M NaOH solution over the course of 3 h; (ii) washing with HCl and water; and (iii) drying at 60 °C for 24 h. Over four consecutive cycles, adsorption decreased in a continuous form from 43 mg/g to near 36 mg/g and at the same time, the desorption efficiency decreased from 71% to 60% after the fourth cycle.

ZnO–CdWO_4_ nanoparticles had been synthesized by a green method with lemon leaf extract to favorably anchor functional groups on their surfaces [13]. The prepared nanoparticles were used as an adsorbent of Congo red. The adsorption process was found to be exothermic and spontaneous, and followed the Freundlich isotherm model. The Boyd plot had been used as a confirmatory tool to fit the adsorption kinetics data alongside both the intraparticle diffusion and pseudo-second-order models. No desorption data were included in this work.

In the next reference [14], the investigation was not carried out on an industrial wastewater but on a synthetic solution of water containing the organic dye, which they specified in the abstract. The metal–organic framework Fe-MIL-88NH_2_, with the composition (Fe_3_O(OH_2_)_3_Cl(NH_2_-BDC)_3_· 9.5H_2_O), was used to remove Congo red (5–60 mg/L) from water. The Temkin adsorption isotherm model and pseudo-second kinetic model best-fitted the experimental data. The removal of the dye from the water responded to a chemical adsorption process. This adsorbent presented a certain grade of selectivity with respect to Congo red and in contrast to both methyl orange and methylene blue. Desorption of the loaded Congo red was done using ethanol and after four cycles the removal efficiency was fixed above 83%.

The mixture of chitosan beads cross-linked with glutaraldehyde was used to investigate the adsorption of reactive blue 4 from an aqueous solution [15]. The response surface methodology was applied to evaluate the concentration of chitosan, glutaraldehyde, and sodium hydroxide on the swelling degree in the adsorbent. The design with r^2^ equal to 0.8634 allowed for to the best concentrations including: chitosan (3.3% *w*/*v*), glutaraldehyde (1.7% *v*/*v*), and NaOH (1.3 M). However, for practical purposes aiming to decrease the viscosity and facilitate the formation of the beads, the real chitosan concentration used in the experiments was equal to 3.0%. In the case of the dye adsorption, the best conditions (r^2^ = 0.8280) were: pH 2, adsorbent dosage of 0.6 g, and initial dye concentration of 5 mg/L. The adsorption process was controlled for mainly by chemisorption interactions and responded well to the Elovich model, expressed as
(4)$${\left[\mathrm{RB}4\right]}_{\mathrm{ad},e}=\frac{1}{\beta }\ln\alpha \beta +\frac{1}{\beta }\ln t$$

And the Freundlich isotherm. In Equation (4), α represents the initial adsorption rate and β represents the desorption constant; this last parameter also corresponds to the extent of the surface coverage and activation energy for a chemisorption mechanism. No desorption data were included in the investigation.

The adsorption of methyl orange and fluorescein sodium was investigated using three 3D coordination polymers: A, (CPs) and [Cd(L)(bifu)]_n_; B: [Zn(L)(bimb)_0.5_]_n_; and C: {[Ni(L)(bifu)](H_2_O)_4_}, being H2L 4,4′-(phenylazanediy1)dibenzoic acid, bifu: 2,7-bis(1-imidazoly)fluorene, and bimb: 1,4-bis(imidazol-l-yl)) [16]. The 3D network structures were different considering A crystallized in a monoclinic space group P21/n; B crystallized in a orthorhombic space group Pbca; and C crystallized in a triclinic space group P-1. The adsorbents were not suitable for the adsorption of methylene blue and rhodamine B. No desorption data were included in this work.

A series of enantiomorphic Pb-based metal–organic coordination polymers, i.e., [Pb((R,R)-TBA)(H_2_O)]·1.7H_2_O (R1) and [Pb((S,S)-TBA)(H_2_O)]·1.7H_2_O (S1) (TBA = 1,3,5-triazin-2(1H)-one-4,6-bis(alanyl)), had been synthesized through the solvothermal reactions [17]. Compounds R1 and S1 crystallized in the chiral space group, P212121, revealing infinite unidimensional (1D) chain structures. These complexes showed a selective adsorption of anionic organic dyes (e.g., Congo red) via a host–guest hydrogen bonding interaction, electrostatic attraction, and π–π stacking between the Pb complex and the organic dye. Dye uptake reached 140 mg/g and followed the pseudo-second order kinetic model. No desorption data were included in this work.

Methyl blue was removed from water by ordered mesoporous silica SBA-15 [18]. The adsorbent was synthesized by silica functionalization with β-cyclodextrin via amide linkage. The adsorption properties of this adsorbent were investigated under several variables: contact time, pH, ionic strength, temperature, and salt. The maximum dye adsorption capacity was of 1791 mg/g, as the adsorption followed the non-linear forms of both the Langmuir and Freundlich isotherms; thus, the dye uptake was attributed both to homogeneous and heterogeneous mechanisms. The organic dye uptake presented a slight decrease in the 2–5 pH range as well as an important decrease from neutral and alkaline values. The adsorption capacity towards the dye was explained in terms of the tailored host–guest interaction between the β-cyclodextrin cavity and the aromatic moiety of the dye, as well as in combination with the electrostatic attraction between amine groups and the sulfonated group present in the organic dye. Ethanol was used to desorb the dye and there was a continuous decrease in the uptake capacity, specifically up to six cycles.

The ternary Cu/Cu_2_O/C composite, presenting anionic dye adsorption capacities, was synthesized by a solvent-free method [19]. The maximum monolayer adsorption capacity of Cu/Cu_2_O/C for methyl orange and acid blue 93 was 1409 and 5385 mg/g, respectively. In addition, the Cu/Cu_2_O/C composite could selectively adsorb anionic dyes (methyl orange and acid blue 93) from a mixture of different dyes. The increase in the adsorbent dosage allowed to achieve the equilibrium at shorter reaction times, whereas maximum uptakes (99%) were reached at pH 4.7, decreasing this value until 50% at pH 11. The various experimental data indicated that the adsorption process followed the Langmuir isotherm and the pseudo-second-order kinetic model, whereas the removal of both dyes from the water was spontaneous and exothermic in nature. Based on the experimental results and the characterization of the composite, it could be concluded that a physical and chemical synergism effect existed in the adsorption process, and chemisorption was the main rate-limiting step. Desorption was achieved using hot water (90 °C) and ethanol after three cycles, and the removal rate decreased from 99% to 75%; this decrease was attributable to (i) the partial oxidation of Cu_2_O or Cu on the surface of the composite and the fact that (ii) the desorption was not complete. 

Combustion and gel calcination operations were used to prepare magnetic Zn_0.3_ Cu_0.7_ Fe_2_ O_4_ nanoparticles (50–80 nm) and adsorb reactive red 2BF [20]. Experimental data using dye concentrations in the 100–400 mg/L range fitted to the non-linear pseudo-second-order kinetic model, whereas the Temkin isotherm explained the experimental loading equilibrium data; thus, adsorption was due to a monolayer–multilayer hybrid adsorption mechanism. The dye adsorption capacity reached 135 mg/g, though it was pH-dependant, with a maximum adsorption in the 1–3 pH range and a progressive decay near the zero adsorption at pH 11. Desorption can be carried out under alkaline conditions, although a continuous decrease in the adsorption rate, from 100% to near 80%, was observed after ten cycles. This decrease was attributable to the nanoparticles agglomerated and the adsorption sites gradually decreased, in addition to the gradual loss of the adsorbent material.

A polyurethane/polyaniline material that can be used as an adsorbent for azo dyes from aqueous solutions was developed [21]. This adsorbent was generated by incorporating polyaniline into the surface of a macroporous polyurethane foam by a chemical polymerization methodology. The adsorption capacity of the material was evaluated against the removal of methyl orange at various pH values, contact times, and initial concentrations. The adsorption reached an equilibrium after 30 h, following the Elovich and intraparticular diffusion models. The maximum experimental adsorption capacity was of 255 mg/g at pH 5, whereas the data fitted the Sips isotherm:(5)$${\left[\mathrm{MO}\right]}_{\mathrm{ad},e}=\frac{{\left[\mathrm{MO}\right]}_{\mathrm{ad},m}K_{S}{\left[\mathrm{MO}\right]}_{\mathrm{aq},e}^{\mathrm{ns}}}{1+K_{S}{\left[\mathrm{MO}\right]}_{\mathrm{aq},e}^{\mathrm{ns}}}$$
where K_S_ represents the equilibrium constant and ns represents a surface heterogeneity parameter. Desorption was investigated using 0.1 M HCl solutions; thus, due to the electrostatic repulsion, the dye was removed from the adsorbent. After ten cycles, the adsorbent maintained its desorption performance, though the adsorption rate decreased from 100% to 60% and this decrease was attributed to loss of adsorbent due to agitation and material management. 

The one-step hydrothermal method allowed for the production of flower-like Mg/Fe-layered double-oxide nanospheres and the adsorbent was used to decrease methyl blue and Congo red levels in aqueous solutions [22]. The removal of 99% of the initial dye concentration occurred quickly in the case of methyl blue, requiring 5 min, while the case of Congo red required 25 min. Following the Langmuir isotherm, in the 20–40 °C range, maximum removal capacities (25 °C) varied from 1250 mg/g in the case of Congo red and 2000 mg/g for methyl blue; these results were attributable to the chemical adsorption between both anionic dyes and cationic adsorbent nanospheres through strong electrostatic interactions. The adsorption of both dyes followed the pseudo-second-order kinetic model in the endothermic and spontaneous adsorption processes. In the case of methyl blue, desorption was done by water (three washes) and the adsorption efficiency (above 99%) was maintained for five cycles.

Akaganeite nanoparticles (average size 50 nm) were synthesized by using microwaves to remove Congo red from an aqueous phase [23]. Experimental results showed that the adsorption kinetics fitted the non-linear pseudo-second-order equation, whereas adsorption equilibrium values, at various temperatures, generally best-fitted the non-linear Langmuir isotherm, with the maximum adsorption capacity of more than 150 mg/g at pH 5.5. Dye uptake onto the adsorbent was spontaneous and endothermic. No desorption data were included in this work.

A composite material of magnetic cross-linked chitosan-glyoxal/ZnO/Fe_3_O_4_ nanoparticles (CS-G/ZnO/Fe_3_O_4_ NPs) was used in the adsorption of reactive blue 19 [24]. The compound was firstly produced by first Schiff’s base magnetic cross-linked chitosan-glyoxal/Fe_3_O_4_ composite (CS-G/Fe_3_O_4_) synthesis and then further by the loading of zinc oxide nanoparticles into the polymeric matrix. Using dye concentrations in the 50–300 mg/L range, the adsorption results were well-described by the pseudo-second-order kinetic model and Freundlich isotherm, with the maximum adsorption capacity of 363 mg/g at 60 °C. Dye uptake was attributable to several types of interactions including (i) electrostatic attractions, (ii) hydrogen bonding, and (iii) n-π interactions. Again, the desorption step was not considered in this work. 

A type of hierarchical porous zeolitic imidazole framework −67@ layered double-hydroxide, ZIF-67@LDH, was developed; the material consisted of nanocubes decorated with nanosheets [25]. This material was used in the adsorption of methyl orange and alizarin red S at various pH values, temperatures, and contact times. Adsorption equilibrium was attained within 100 min at 30 °C, with a slight increase of the adsorption capacity with the increase of the temperature (20–40 °C). The adsorption followed the pseudo-second-order kinetic model and Langmuir isotherm, which confirmed the monolayer adsorption of the dye onto the adsorbent. Maximum capacities were 793 mg/g at pH 4 (alizarin red S) and 316 mg/g at pH 6 (methyl orange). Desorption was accomplished by the use of 0.1 M of HCl solution, followed by a second wash with ethanol and water. After four cycles, the adsorbent presented a continuous decrease in its adsorption capacity. 

The material, Ag@ZIF-67, was used in the removal of methyl orange from polluted water [26]. Dye uptake increased from pH 2 to pH 6, with a further decrease until pH 11. This uptake also increased with temperature in the 10–30 °C range, with no variation at 40 °C. The experimental data fitted to the pseudo-second-order kinetic and Langmuir models, with the maximum adsorption capacity of 1146 mg/g. Dye desorption used 0.1 M of HCl solution and a continuous decrease in the adsorption–desorption capacity was observed, e.g., 99% adsorption (first cycle) against 70–80% (fifth cycle). 

As pristine carbon materials poorly remove organic dyes from waters, tuning the carbon with magnetic compounds tended to reverse the above situation. Thus, carbon biomass from the seeds of Moringa oleifera was mixed with manganese ferrite and the final material was investigated to adsorb indigo carmine, acid blue 158, and reactive blue 4 [27]. Dye uptake onto the adsorbent was 34 mg/g, 35 mg/g, and 32 mg/g for the above dyes, respectively, with the maximum adsorption occurring at a pH near 3. In addition, in the three cases, the adsorption followed the pseudo-second-order model and Freundlich isotherm in the endothermic and spontaneous processes. In the presence of co-existing anions, chloride, sulphate, or bicarbonate, the adsorption slightly decreased, this attributable to the competition with the dye molecules for vacant sites of the adsorbent. The adsorbent worked via electrostatic attraction, hydrogen bonding, and π–π interaction. Desorption was performed using 0.1 M of NaOH solution, with a relationship of 50 mL desorption solution/100 mg of adsorbent loaded with dye. After five cycles, the efficacy of the adsorbent decreased, probably due to the increasing clustering of the adsorbent surface. Some of these results are shown in Table 2.

Alginate-reinforced reduced graphene oxide@hydroxyapatite hybrids have been synthesized via the co-precipitation method and investigated as adsorbents towards reactive blue 4, acid blue 158, and indigo carmine from aqueous solutions [28]. The corresponding uptake was of 46 mg/g, 48 mg/g, and 47 mg/g, respectively. Within this adsorbent, the pH value to generate the maximum adsorption was 6–7 and the dyes removal was both endothermic and spontaneous. The presence of bicarbonate in the solution did not affect the adsorption properties; however, if chloride or sulphate anions were present in the water, the adsorption efficiency would decrease due to interactions with the adsorbent surface. Dyes’ uptake was attributable to (i) electrostatic interactions, (ii) surface complexation, and (iii) hydrogen bonding mechanisms. Desorption was carried out with 0.1 M of NaOH solutions, decreasing the adsorption capacity from the first to the fifth cycle in a continuous form. 

Zeolitic imidazolate framework-67 was used as a template to fabricate hollow LDH nanocages and the template was consumed during the formation of the nanomaterial. The nanocages were composed of NiCo-LDH nanosheets that maintained the dodecahedron shape of the template [29]. The resultant material was used to remove Congo red under various experimental variables. Dye adsorption was attributable to a chemisorption process and followed the pseudo-second-order kinetic model and Langmuir isotherm in the 25–45 °C temperature range. With a maximum adsorption at pH 7, the maximum capacity increased with the increase of the temperature, e.g., 926 mg/g (25 °C) or 1197 mg/g (60 °C). The nanocages could be regenerated through the solvothermal method using methanol and treated at 80 °C over the course of 12 h. Continuous use of the adsorbent demonstrated that there was a slight but continuous loss of the adsorptive capacity for up to five cycles.

The cyclomatrix phosphazenes-co-3,3′-sulfonyldianiline PSD-NH_2_ microspheres were used to remove methyl orange from solutions [30]. As it was usual in this type of investigation, several variables were considered in the study including: pH, adsorbent dosage, contact time, initial dye concentration, and temperature. The experimental data indicated that the adsorption capacity of the microspheres was 187 mg/g under the optimal adsorption conditions and followed the pseudo-second-order kinetic and Langmuir isotherm models. Congo red was adsorbed due to (i) intermolecular electrostatic interaction; (ii) hydrogen bonding between the microspheres adsorbed on the surface and the dye; and (iii) π–π and C-H…π stacking, in addition to other molecular forces. Dye uptake was endothermic and spontaneous in nature. Desorption data were not included in this work.

Attapulgite modified with 3-aminopropyltriethoxysilane was used in the adsorption of Congo red from waters [31]. It was described that the pseudo-second-order equation and Sips model best-fitted the experimental data. Contrary to the attapulgite behavior, the adsorption via the modified attapulgite was affected neither by the increase of the temperature, nor by the change in the pH value. Desorption was not considered here.

## 3. Adsorbents for the Removal of Cationic Dyes from Waters

Novel magnetic MgFe_2_O_4_/crumbled reduced GO nanoparticles with a surface area of 35 m^2^/g and adsorption capacity near 25 mg/g were used to eliminate methylene blue (MB) from solutions [32]. In the investigation, several experimental variables were considered: adsorbent dose, dye concentration, pH, rate of agitation, and the change in temperature. The elimination of the dye followed a pseudo-second-order reaction, expressed as
(6)$$\frac{t}{{\left[\mathrm{MB}\right]}_{\mathrm{ad},t}}=\frac{1}{K_{p2}{\left[\mathrm{MB}\right]}_{\mathrm{ad},e}^{2}}+\frac{t}{{\left[\mathrm{MB}\right]}_{\mathrm{ad},e}}$$

And characterized by multi-layer formation on the solid surface. In the above equation, [MB]_ad,e_ and [MB]_ad,e_ are the dye concentrations in the adsorbent at the equilibrium and at an elapsed time, respectively; K_p2_ is the rate constant; and t is the elapsed time. The values of the enthalpy (endothermic process) and entropy change (219 J/mol K) indicated a strong interaction between the dye molecules and adsorbent surface, as well as an increase of the random arrangement between the organic dye and solid surface. The 10^−3^ M NaOH solution was used as the desorbent. This magnetic separable adsorbent retained 75% of its initial adsorption capacity after four consecutive cycles. 

A super-adsorbent nanocomposite hydrogel adsorbent was prepared by employing vinyl hybrid silica nanoparticles as a cross-linking agent [33]. This adsorbent presented a remarkable adsorption capacity (1690 mg/g) towards methylene blue, with a high removal ratio (90%) within 40 min. To recover the dye from the loaded adsorbent, 0.1 M HCl solution was used. After five cycles, there was a continuous decrease from 899 mg/g to 757 mg/g in the adsorption capacity.

Used as precursors, 1D [Mo_3_O_10_]_n_^2n−^ chains and the flexible N,N′-bis(5-methyl-2-pyrazinecarboxamide)-1,2-ethane (bmpae), a 3D metal–organic framework ([Cu(bmpae)0.5(Mo_3_O_10_)(H_2_O)]·H_2_O), had been hydrothermally synthesized [34]. Its adsorption capacities towards gentian violet and methylene blue were investigated. It was mentioned in the manuscript that acetonitrile was used in the desorption step.

The superparamagnetic Mn-metal organic framework (MOF) with core-shell nanostructures, synthesized in situ, was developed as a magnetic adsorbent for malachite green removal from aqueous solutions [35]. Three variables were investigated: pH, contact time, and agitation speed. The results showed that the CoFe_2_O_4_@Mn-MOF core-shell removes about 98% of the dye from a water medium. This reference seemed to be a rare exception amongst the many published adsorption studies in which the agitation speed was considered an experimental variable to investigate. The effect of this variable on malachite green adsorption was investigated in the 100–400 min^−1^ range and the results showed that maximum adsorption (99%) was reached at 200 min^−1^; this result was attributable to the decrease of the film boundary layer that wrapped particles and thus adsorption increased. No desorption data were included in the investigation.

It was reported [36] that the preparation of carboxylic acid-modified polysilsesquioxane aerogels via a straightforward acid-base catalyzed the sol-gel approach by using methyltrimethoxysilane (MTMS) and the novel and stable 5-(trimethoxysilyl)pentanoic acid. Besides the adsorption of heavy metals (zinc(II) and copper(II)), the adsorbent was used in the removal of methylene blue and rhodamine B. Experimental data fitted well with the Langmuir isotherm, with the maximum adsorption capacities of 154 mg/g (rhodamine B) and 106 mg/g (methylene blue). An increasing content of carboxylic acid groups influenced the morphology, specific surface area, and adsorption behavior of the synthesized aerogels. Desorption was investigated in the case of methylene blue; loaded dye onto the adsorbent was desorbed using HCl solutions of pH 1–2 for 24 h. Under continuous use, the adsorption efficiency was near 92% for the three first cycles but decreased from the fourth cycle, whereas desorption maintained at an 87% for the fourth and fifth cycles. These carboxylic acid-modified aerogels only removed positively charged molecules from mixed-dye solutions.

Hollow polydopamine microcapsules (H-PDA-MCs), synthesized by an oxidation self-polymerization of dopamine using iron trioxide (Fe_2_O_3_) nanocubes, were used to adsorb cationic dyes such as methylene blue [37]. The results showed that when near 25 °C, the maximum adsorption capacity was 192 mg/g, with the equilibrium reached within 1 h. The data fitted to the pseudo-second-order kinetic model, the Langmuir model and the Temkin isotherm:(7)$${\left[\mathrm{MB}\right]}_{\mathrm{ad},e}=B_{1}\ln K_{t}+B_{1}\ln{\left[\mathrm{MB}\right]}_{\mathrm{aq},e}$$
where B_1_ represents the Temkin constant associated with the adsorption heat and K_T_ was the equilibrium binding constant. The increase of the temperature increased the dye uptake onto the adsorbent (Table 3) and thus the adsorption was an endothermic and spontaneous process. In this system, desorption was performed with 1 M of HCl solution and sonication for 60 min, and after a post-treatment the adsorbent was reused.

Hierarchical microspheres of ZnOHF were produced using DL-alanine as the structure-directing agent [38]. These microspheres were formed by numerous nanotubes and were used in the adsorption of organic dyes, though the adsorption of cationic dyes was favored with respect to neutral or anionic dyes. Adsorption data fitted to the Langmuir isotherm and pseudo-second-order kinetic model. The maximum adsorption capacity of the microspheres was 140 mg/g for malachite green at 25 °C in an endothermic and spontaneous physical adsorption process. After five cycles, adsorption decreased from 100% to 75%, though it was not indicated if the dye was desorbed or the adsorbent was used without desorption. 

Mesoporous silica derived from iron ore tailings was synthesized [39] and used in the adsorption of methylene blue. The removal of the dye (pH 10) responded to the Langmuir isotherm and pseudo-second-order equation, and thus to a monolayer adsorption, with the adsorption capacity of 192 mg/g. The desorption step was not considered here. 

Methylene blue and basic violet 16 dyes were removed from an aqueous solution using sulfonated poly(ether ether ketone) (sPEEK) [40]. Kinetic experiments revealed that in the case of both dyes, data were well-fitted by the pseudo-second-order kinetic model, whereas the required times to achieve equilibrium were determined as 40 and 20 min for methylene blue and basic violet 16, respectively. Maximum adsorptions occurred at the pH of 3–6 (methylene blue) and 3 (basic violet 16). The adsorptions responded to the Langmuir model and the maximum adsorption capacities were found to be 98 mg/g (methylene blue) and 182 mg/g (basic violet 16). In the case of methylene blue, the adsorption process presented an endothermic and spontaneous character. This dye was desorbed using 3 M nitric acid/ethanol (30/70% *v*/*v*) mixtures for 20 min and recycling experiments on methylene blue showed that the adsorption efficiency decreased from 99.6% to 99.2% after five cycles, whereas low desorption percentage values of basic violet 16 indicated that this adsorbent may be used for immobilizing the dye.

A natural adsorbent of graham flour was selected as an adsorbent to remove cationic crystal violet from water [41]. The effective pH range for dye adsorption was in the neutral pH region, following the Langmuir isotherm, with the maximum adsorption capacity of 162 mg/g. The polymeric GF adsorbent was selective respect to crystal violet from the synthetic polluted sample, even in the presence of a high concentration of diverse competing ions. The adsorbed dye could be desorbed with ethanol, with a continuous decrease in its capacity observed for up to seventh cycles. 

The next reference deals with the encapsulation of methylene blue on the polymeric natural carbohydrate of the turmeric powder adsorbent [42]. The maximum dye removal of 99.5% was obtained at pH 7. In the acidic pH region, the positively charged protonated adsorbent did not favor the adsorption of positively charged protonated methylene blue due to electrostatic repulsion. In the neutral pH area, the removal of the dye from the solution was explained by electrostatic attraction and complexation of the dye with the adsorbent. The adsorption data were fitted to the Langmuir isotherm, with a maximum adsorption capacity of 157 mg/g. Ethanol can be used to desorb the dye from the adsorbent; its original capacities were reduced for up to seventh cycles. 

Sulfonic acid-functionalized carbon nanotubes’ (CNT-FA-SA) composite was synthesized by Diels–Alder (DA) chemistry and the electrophilic substitution reaction [43], and the adsorption of methylene blue onto CNT-FA-SA was investigated. The loading capacity, in an exothermic adsorption process, increased with the increase of the pH from 2 to 12. The non-linear fitting of the data to Freundlich or Langmuir isotherm models estimated that the maximum adsorption capacity of the monolayer was 1584 mg/g. The adsorption kinetic responded to the non-linear form of the pseudo-second-order equation. The data indicated that the adsorption process was related to chemisorption and physical adsorption simultaneously, and that the adsorption occurred on a heterogeneous multilayer surface. No desorption data were included in the manuscript. 

A trichlorotriazine-derived calix[4]arene and 1,3,5-tris(4-aminophenyl) benzene were cross-linked to produce a calix[4]arene cross-linked polymer with adsorptive properties towards cationic dyes from water [44]. This polymer showed the removal efficiency as over 99.6% and 99.4% for methylene blue and toluidine blue within 5 min, respectively, and Table 4 presents some of the equilibrium capacities obtained with this adsorbent. In the case of methylene blue and toluidine B, the adsorption followed the pseudo-second-order model and the Langmuir isotherm. Maximum equilibrium adsorption capacities for methylene blue and toluidine blue were 1807 and 2161 mg/g, respectively. These results were attributable to electrostatic interactions, π−π interactions, and the hydrogen bonding between the polymer and organic dyes. In column experiments, the removal of the above dyes was of 99%, except in the case of sodium fluorescein which reached 80%. Desorption was investigated in the column operation and by using methanol as a desorbent; after five cycles, the removal efficiency was greater than 95%.

A hexatungstate derivative, [Ni_2_(L)_3_]_2_[W_6_O_19_]·2H_2_O (HL: 2-acetylpyrazine-N(4)-methyl thiosemicarbazone), has been synthesized, characterized, and used in the adsorption of methylene blue, gentian violet, and fuchsin basic [45], with removal efficiencies of 99%, 94%, and 94%, respectively. The different reactivity over the various dyes was caused by the electrostatic interactions between the dyes and the [W_6_O_19_]^2−^ molecules. Methylene blue uptake onto the adsorbent followed the pseudo-second-order kinetic equation. After five cycles of adsorption–desorption, the capacity decreased from 99% to 82%, this decrease attributable to the loss of adsorbent during each washing step.

Rice husk ash can be used as a sustainable source of silica for producing high valuable subproducts. This product served as a precursor for mesostructural graphene oxide and was used to adsorb methylene blue from solutions via oxygen functional groups presented in the adsorbent [46]. The adsorption capacity depended on the gelation pH, graphene oxide content, adsorbent dosage, and initial dye concentration. The highest adsorption capacity was 633 mg/g and the data fitted to the Freundlich or Langmuir models depending of the type of adsorbent material used in the investigation. In all the cases, the data fitted to the pseudo-second-order kinetic model. Desorption information was not included in the manuscript.

HKUST-1/cellulose/chitosan aerogel composite with hierarchical pores was investigated to remove methylene blue [47]. With a high adsorption capacity (526 mg/g), the removal of the dye increased from pH 3 to 7, with levels off until pH 8 and then decreasing from pH 9 to 11. It was determined that the adsorption occurred through H-bond interactions between oxygen and nitrogen atoms in the composite aerogel and nitrogen atoms from the organic dye. The adsorption fitted to the pseudo-first and pseudo-second-order models, and thus the adsorption was due to a synergistic effect of chemical and physical mechanisms. Desorption was investigated using ethanol and water under lyophilization. After five cycles, the adsorption capacity decreased from 162 mg/g to 153 mg/g.

The adsorption behavior of different adsorbents has been studied by various research works but few studies have compared linear and non-linear isotherm and kinetic models alongside phenomenological coefficients. The effect of activated carbon black on methylene blue adsorption behavior of alginate was examined [48]. A low-cost and green adsorbent was fabricated to easily detach from water. The results were well-fitted with non-linear pseudo-second-order and linear Langmuir models. The intraparticle diffusion model and phenomenological coefficients represented the control of adsorption by film diffusion and its limiting by pore diffusion. No desorption data were included in this work.

A graphitic carbon nitride was developed to decrease the contamination produced by diverse compounds, including dyes [49]. The investigation was focused on the removal of methylene blue from aqueous solutions. The dye was removed from the solution very quickly, in a matter of seconds, without any additional physical or chemical activation. No desorption data were included in this work.

The next investigation showed the abilities of F-terminated Ti_3_C_2_T_x_-MXene as an adsorbent of methylene blue-bearing wastewater [50]. The adsorption efficiency towards this dye sharply decreased from pH 4 to pH 7, with maximum adsorption achieved at pH 2. The adsorption was based on the following reactions:(8)$${\left[\mathrm{Ti}-F\right]}^{-}+{\mathrm{MB}}^{+}=\mathrm{Ti}-F-\mathrm{MB}$$
(9)$${\left[\mathrm{Ti}-O\right]}^{-}H^{+}={\left[\mathrm{Ti}-O\right]}^{-}+H^{+}$$
(10)$${\left[\mathrm{Ti}-O\right]}^{-}+{\mathrm{MB}}^{+}=\mathrm{Ti}-O-\mathrm{MB}$$

Thus, F and O terminations attracted methylene blue (MB) molecules. Other dyes (methyl orange and rhodamine B) could be also adsorbed, though the adsorption was dependent on the exposure time, increasing with the time in the case of the above dyes. The removal efficiency of methylene blue slightly decreased from the first to the fourth cycle.

A material formed by polystyrene sulfonate on a laterite soil was investigated to adsorb crystal violet and methylene blue from an aqueous solution [51]. Both dyes were effectively removed from the solution after 75 min at pH values of 8 (crystal violet) or 9 (methylene blue). In addition, in both cases, the adsorption efficiency increased from pH 2 to alkaline values. The adsorption data derived from both dyes indicated exothermic and spontaneous adsorption processes, and fitted-well to the Langmuir model, with the capacity in a solution of ionic strength 1mM of about 43 mg/g (crystal violet) and 14 mg/g (methylene blue). Desorption data were absent in the investigation.

A natural bentonite was used as an adsorbent to remove basic red 46 from an aqueous solution [52]. Throughout the experimentation, an adsorbent concentration of 0.1 g/L was used. At pH 7 and 25 °C, the maximum adsorption capacity was 594 mg/g, with the experimental data best-fitted to the Langmuir isotherm and pseudo-second-order kinetic model. The increase of the temperature was associated to a decrease of dye uptake onto the adsorbent. Investigations about the desorption step were not included in this work.

As it was mentioned above, pristine carbon materials tended to have a low dye-removal capacity. One method to improve this fault was to tune the carbonaceous material with a complementary one. One type of material that could be added was biomaterials. In the next reference, a sulfur-tethered adsorbent of the chitosan-tapioca peel biochar composite was fabricated. The peel of tapioca acted as a precursor for the fabrication of the biochar matrix. The removal of malachite green and rhodamine B from solutions using this adsorbent was investigated [53]. Using dye concentrations of 50 mg/L, the uptakes were 53 mg/g (malachite green) and 41 mg/g (rhodamine B) at a pH near 8 and after 120 min. The adsorption followed the pseudo-second-order kinetics and Langmuir isotherm equations. As the norm, dye adsorption was attributable to (i) electrostatic attractions, (ii) hydrogen bonding, and (iii) π–π interactions. Using 0.1 M of NaOH solutions to desorb the loaded dyes, the adsorbent was used in five successive cycles of adsorption–desorption with a continuous decrease in its properties to remove the dyes from the solution.

In a very similar investigation to the one above [54], the same adsorbent and dyes were used, though in this case [53], the dyes’ concentrations were 25 mg/L because the uptake was lower than in the previous reference, namely 30 mg/L versus 53 mg/g (malachite green) or 33 mg/g versus 41 mg/g (rhodamine B). Additionally, the best uptakes were yielded at a pH near 8 and after 120 min of reaction. The experimental data also followed the pseudo-second-order kinetic model, but in this case, the adsorption fitted to the Freundlich isotherm instead of the Langmuir isotherm. The adsorbent also lost its properties under the five cycles of adsorption–desorption.

Zn_3_(OH)_2_V_2_O_7_·2H_2_O nanocables have been fabricated by a DL-Alanine-assisted hydrothermal route and were used in the adsorption of methylene blue [55]. The maximum adsorption capacity was 113 mg/g at 25 °C in a physisorption process, following the pseudo-second-order equation and Langmuir isotherm. No desorption data were included in this work.

A mixed process of biomodification and magnetic treatment was used to prepare polydopamine-modified graphene-based adsorbents, which incorporated active Fe_3_O_4_ nanoparticles (average size of 6.5 nm) [56]. With an increase of the adsorption capacity from pH 2 to 10, the adsorption equilibrium was reached after 7 h. The adsorption of methylene blue followed the pseudo-second-order kinetic model, the intraparticle diffusion model, and the Langmuir isotherm. Maximum adsorption capacities were 132 mg/g (30 °C), 140 mg/g (40 °C), and 152 mg/g (50 °C), with the adsorption presented as endothermic and spontaneous. It also showed an increase in the randomness at the solid-solution interface. Desorption followed an acid treatment to induce the dye removal from the adsorbent, this caused by the generation of an excess of protons which competed with the cationic dye and endowed the amine groups of the adsorbent with a positive charge with the subsequent repulsion of the dye molecule. After five cycles, the adsorbent showed a continuous decrease of its initial removal rate.

Another adsorbent-based metal–organic framework with nanometer blocks of amide-functionalized Fe(III)-based metal–organic frameworks were produced via the ultrasonic method and without the addition of any surfactants, at room temperature and with atmospheric pressure [57]. It was used in the removal of methylene blue for the solution. No desorption data were included in this work.

A β-cyclodextrin/graphene oxide composite was prepared by modifying graphene oxide via the β-cyclodextrin cross-linking method [58]. The composite was used to investigate its possibilities in the removal of methylene blue from an aqueous solution. Under the optimal conditions of temperature (70 °C), time (60 min), and adsorbent dosage (0.04 g/L), and at pH 7, a removal efficiency of 90% was achieved, which compared better than that of the graphene oxide (70%). The maximum adsorption capacity was 76 mg/g and the dye was removed from the adsorbent by desorption with absolute alcohol during 12 h. Removal efficiency was maintained at about 90% after nine cycles.

As *physalis alkekengi* L. husk is an agricultural waste product, its disposal causes environmental pollution of water and air. Thus, in order to give a further utility to this waste, it was used as precursor of a porous carbon material. Malachite green (MG) was used to investigate the adsorption properties of the material [59]. It was found that at a pH of 9, the maximum adsorption capacity of the adsorbent reached near 2000 mg/g, whereas 51% (1016 mg/g) of the dye was adsorbed in the first 5 min. Dye uptake followed the pseudo-second-order and Banghman models
(11)$${\left[\mathrm{MG}\right]}_{\mathrm{ad},t}={\left[\mathrm{MG}\right]}_{\mathrm{ad},e}\left(1-e^{-K_{4}t^{n}}\right)$$
where K_4_ represents the Bangham rate constant and n represents the power of the Bangham model. The increase of the temperature in the 20–40 °C range produced a slight increase in the adsorption capacity, which followed the Langmuir model. Deionized water was used to desorb the dye and the adsorbent was recarbonized at 600 °C under nitrogen atmosphere. After ten cycles, the adsorbent decreased its removal rate from 100% to 89%, though further recycling caused a slight decrease in the adsorption capacity, probably caused by the deposition of the byproducts on the adsorbent surface.

## 4. Adsorbents for the Removal of Anionic and Cationic Dyes from Waters

A magnetic metal–organic framework adsorbent (Fe_3_O_4_@UiO-66), based on functionalized magnetic Fe_3_O_4_ nanoparticles and highly water-stable UiO-66, was used to remove methyl orange and methylene blue from aqueous solutions [60]. In the case of methylene blue, the adsorption data fitted well to the linearized Freundlich model:(12)$$\ln{\left[\mathrm{MB}\right]}_{\mathrm{ad},e}=\ln K_{F}+\frac{1}{n}\ln{\left[\mathrm{MB}\right]}_{\mathrm{aq},e}$$
where K_F_ represents the constant model, [MB]_aq,e_ represents the methylene blue concentration in the solution at the equilibrium, and n_F_ represents a value which indicates whether the process is favorable (n_F_ greater than one) or not. The experimental data derived from the adsorption of methyl orange (MO) indicated that the linearized form of the Langmuir isotherm was the best fit.
(13)$$\frac{{\left[\mathrm{MO}\right]}_{\mathrm{aq},e}}{{\left[\mathrm{MO}\right]}_{\mathrm{ad},e}}=\frac{1}{K_{L}{\left[\mathrm{MO}\right]}_{\mathrm{ad},m}}+\frac{1}{{\left[\mathrm{MO}\right]}_{\mathrm{ad},m}}{\left[\mathrm{MO}\right]}_{\mathrm{aq},e}$$

In this equation, K_L_ represents the model constant and the subscript ad,m represents the maximum dye concentration in the adsorbent. This adsorbent had adsorption uptakes of 244 mg/g and 205 mg/g for methyl orange and methylene blue, respectively. The adsorption kinetic experiments demonstrated that the adsorption data for both dyes well-fitted the pseudo-second-order model. The thermodynamic data indicated that the adsorption of both dyes onto the adsorbent was spontaneous and presented an endothermic character. To release the dyes loaded onto the adsorbent, water and ethanol under ultrasonic treatment were used. After eight cycles, there was a continuous decrease in the adsorption capacity of both dyes onto the adsorbent.

Two Zr-based metal–organic cages, namely Zr-MOC-1 (tetrahedral structure) and Zr-MOC-2 (capsule-like cage skeleton), were synthesized under solvothermal conditions [61]. The adsorption performances of Zr-MOCs for the organic dyes (methylene blue, crystal violet, methyl orange, and orange G) from aqueous solutions were reported. The desorption step using ethanol was only mentioned in the case of methylene blue.

An iron-based Fe-BTC MOF, prepared according to an aqueous-based procedure, was used as an adsorbent for the removal of alizarin red S and malachite green from aqueous phases [62]. Optimal adsorption was reached at pH 4 due to the favorable interactions between the dyes and the adsorbent. The kinetic investigation concluded that for the two investigated dyes, the pseudo-second-order model was followed. Accordingly with the Langmuir model, maximum adsorption capacities of 80 mg/g and 177 mg/g were found for alizarin red S and malachite green, respectively. No desorption data were included in the published manuscript.

The solvothermal self-assembly between Cd^2+^ and a hexacarboxylic acid created the porous material formulae as [(CH_3_)_2_NH_2_]_6_[Cd_3_(L)_2_]·5DMF·3H_2_O (H_6_L responded to 3,4-di(3,5-dicarboxyphenyl)phthalic acid), which selectively adsorbed cationic dyes against neutral or anionic compounds [63]. The investigation was based on the adsorption of azure A+ and methylene blue+, with adsorption capacities of 698 mg/g (azure A+) and 573 mg/g (methylene blue +). As acetonitrile cannot desorb the dye, desorption was performed using a NaCl-acetonitrile solution and the mission of Na^+^ ions was to displace the dye cation to conserve the stability of the adsorbent skeleton and charge balance. After five cycles, both adsorption and desorption capacities slightly but continuously decreased.

A carboxyl-functionalized covalent organic framework (TpPa-COOH) was synthesized under environmental pressure and the adsorptive properties were investigated using eosin B (anionic dye) as well as the cationic dyes methylene blue and safranine T [64]. TpPa-COOH showed a selective adsorption effect on the two cationic dyes. In these two cases, the adsorption isotherm responded to the Langmuir model, with maximum capacities of 1135 mg/g (safranine T) and 410 mg/g (methylene blue). The desorption step was only investigated in the case of safranine T and the step was carried out with tetrahidrofurane and 1 M HCl; the lean adsorbent was washed with ethanol and dried under vacuum at 60 °C over the course of 12 h. After three cycles, the removal efficiency decreased from 99% to 95%.

Magnetic chitosan hydrogel beads, based on chitosan biopolymer, magnetic graphene oxide nanoparticles (GO/Fe_3_O_4_), and isophthaloyl chloride, were prepared and used in the removal of cationic and anionic dyes from an aqueous solution [65]. Experimental results showed that data were best-fitted to the pseudo-second-order kinetic model and non-linear Langmuir isotherm. The maximum adsorption capacities of methylene blue and eriochrome black T were 289 (pH 8–10) and 292 (pH 3) mg/g, respectively. The removal of the dyes from the solution was associated to an endothermic, spontaneous, and favorable process, as the R_L_ values derived from:(14)$$R_{L}=\frac{1}{1+K_{L}{\left[\mathrm{dye}\right]}_{\mathrm{aq},0}}$$

In which 0.012 (methylene blue) and 0.010 (eriochrome black T) are included. In the above equation, K_L_ represents the Langmuir constant estimated by the corresponding fit and [dye]_aq,0_ represents the initial dye concentration in the aqueous solution. Dye uptake was attributed to the electrostatic interactions of the dyes with the primary amine and hydroxyl and carboxyl groups of the adsorbent beads. Desorption was performed using HCl (pH 3) in the case of methylene blue or NaOH (pH 9) in the case of eriochrome black T. The adsorption–desorption properties were well-maintained for up to four cycles but from the fifth cycle, they began to decrease.

A boron-doped mesoporous carbon material was generated from F127 (block polyether (EO_106_-PO_70_-EO_106_)) as a soft template and both phenolic resin and triisopropyl borate were selected as the carbon and boron-containing precursors [66]. The prepared material had been used in the adsorption of crystal violet and Congo red dyes. In both cases, the experimental data fitted the Langmuir isotherm and the pseudo-second kinetic equation, with dye uptakes of 385 mg/g (crystal violet) or 323 mg/g (Congo red). After five rounds of circulation, the removal rate decreased from 79% to 63% (Congo red) and 94% to 80% (crystal violet); this decrease may be caused by the disappearance or weakening of the vacancy sites on the adsorbent in the desorption process. In the published manuscript, how the desorption procedure was carried out was not clearly mentioned.

The adsorption performance of delaminated Ti_3_C_2_-MXenes towards six different organic dyes in aquatic media at different pH levels and ionic strengths was investigated [67]. The ability of this adsorbent to remove the dyes was based on the electrostatic interactions between the ionizable functional groups of MXenes and the dyes. The pH had a different influence on the dye adsorption and in the case of methylene blue, the adsorption of this dye increased with the pH value from 2 to 10. Methyl orange presented an increase in its adsorption from pH 2 to 4, with a further stabilization until pH 10. The removal of Congo red was stable in the 2–10 pH range and then decreased until pH 12. Methyl red was adsorbed in the range of 2–4 with a sharp decrease up to pH 10 (negligible adsorption). Lastly, in the case of orange G, a 60% adsorption rate was achieved at pH 2 and near zero at pH 4. In water, methyl red (MR) adsorption was attributable to the intraparticle diffusion model:(15)$${\left[\mathrm{MR}\right]}_{\mathrm{ad},t}=K_{p}t^{0.5}+{\left[\mathrm{MR}\right]}_{\mathrm{ad},t=0}$$
where K_p_ represents the rate constant. In all the other dyes, the fitting of the results was not clear. No desorption data were included in this work.

Highly dispersed graphene nanosheets were directly integrated into a polyurethane sponge and the resulting material was used as an adsorbent towards methylene blue, ethidium bromide, and eosin Y [68]. The dye uptake followed the Langmuir isotherm model, which provided evidence of a strong monolayer chemisorption. In the three cases, the adsorption process was spontaneous and exothermic, and the dye uptake fitted to the pseudo-second-order kinetic model. In the case of the cationic dyes, the adsorption increased with the increase of the pH, e.g., the maximum adsorption capacity at pH 10 within methylene blue and ethidium bromide, whereas anionic dyes followed the opposite tendency, e.g., pH 3 for eosin Y. Again, desorption data were absent in this work.

An aerogel formed by graphene-polydopamine-bovine serum albumin, to which biopolymers were added, was used in the removal of organic dyes (methylene blue and Evans blue), toxic metals (Cr(VI) and Pb(II)), and organic solvents (n-hexane,n-heptane and toluene) [69]. In the case of both organic dyes, the adsorption capacity increased when the pH value increased from 2 to 8. Desorption was only investigated in the case of methylene blue using ethanol at pH 2 during 24 h. After three cycles, the removal efficiency for methylene blue decreased by 2–20% (depending on the dye concentration). The system was used in a continuous form using a flow-through filtration system.

Using zirconium, a Zr-based metal–organic framework was synthesized by a solvothermal procedure. The adsorptive characteristics of such materials were investigated against the presence of methyl red and methylene blue in solutions [70]. Experiments showed that a change in the activation solvent and activation method altered the physical properties of the adsorbent (surface area, pore-volume, pore diameter, and surface charge), and thus the capacity to remove the dyes was also altered. Some of the experimental results are summarized in Table 5.

The adsorption of methylene blue increased with the increase of the pH value, with a maximum capacity of 217 mg/g, whereas that of methyl red reached a maximum at pH 2 (243 mg/g); thus, by controlling the pH of the solution, the adsorbent can be used to selectively separate one from the another. In both cases, adsorption followed the Langmuir and pseudo-second-order kinetic models, and the adsorption was both spontaneous and endothermic. Desorption used acidic ethanol solutions under ultrasonic assistance over the course of 1 h as the desorbent procedure. After five cycles, a slight decrease in the adsorption capacity was shown either for methyl red or methylene blue dyes.

Cobalt ferrite nanoparticles capped with an ultrathin phosphate layer of near 1 nm thick were synthesized and characterized using usual techniques [71] and they were used in the adsorption of a number of dyes. These capped nanoparticles adsorbed 4.5 mg/g (89.5% adsorption) of methylene blue within 5 min, a value which compared well with the practically non-adsorption quality of the dye using the uncoated cobalt ferrite. Using solutions of 50 mg/L of the corresponding dye, the adsorption efficiency for various dyes included: brilliant blue R (92%), bromo phenol blue (75%), and methylene blue (91%). No desorption data were included here.

The in-situ growth of zeolitic imidazolate frameworks on zinc-layered double-hydroxide, which developed in a porous composite material (ZIF-8@ZnAl-LDH), was investigated [72]. This material was used in the removal of malachite green and methyl orange from water. In both cases, adsorption equilibrium was reached after 2 h and it followed the Langmuir isotherm. Due to the nature of these dyes, the pH influenced their adsorption in contrary manner: an increase of the pH from 2 to 10 favored the adsorption of malachite green, whereas methyl orange was best removed from the solution at acidic pH values (e.g., 3). Ethanol was effective in the desorption of both dyes and after four consecutive cycles, the adsorbent maintained a 97% and 95% adsorption rate for malachite green and methyl orange, respectively, of its initial adsorption capacity. These results were attributable to the adsorbent characteristics including: (i) greater superficial area, (ii) porous structure, and (iii) exposed metallic nodes within the specific morphological inter-layered structure.

A biosynthesis strategy using MgO nanoparticles and various *Tecoma stans* (L.) plant extracts (flower, bark, and leaf) was developed in order to find a suitable adsorbent for organic dyes [73]. The investigated dyes were Congo red and crystal violet, and the variables investigated were pH, time, and dye concentration. The best conditions for Congo red uptake (99.7%) were a pH of 7.9 and concentration of 9.3 mg/L, whereas crystal violet was adsorbed (90.8%) at a pH value of 6.3 and concentration of 5 mg/L. Maximum adsorption capacities were produced when the adsorbent presented when the flower extract was used and it varied from 89 (Congo red) to 150 mg/g (crystal violet). The adsorption of both dyes followed the Langmuir isotherm and pseudo-second-order kinetic model. Under the continuous use of three cycles, there was a continuous decrease in the effectiveness of both adsorption–desorption operations.

Again using metal–organic frameworks, two mixed compounds, namely {[Zn_2_(5NO_2_-IP)_2_(L)_2_](H_2_O)}_n_ , named ADES-1, and {[Cd_2_(5NO_2_-IP)_2_(L)_2_(H_2_O)_4_](L)(H_2_O)(CH_3_OH)_6_}_n_, named ADES-2, and where 5NO_2_-IP = 5-nitroisophthalate and L = (E)-N′-(pyridin-3-ylmethylene)nicotinohydrazide), had been synthesized and characterized via normal techniques [74]. ADES-1 allowed for the rapid and efficient adsorption of both cationic and anionic dye molecules, though it showed a preference for methyl violet (86% adsorption) if compared to the other dyes that were investigated; some of these results are summarized in Table 6. The usefulness of ADES-1 in the separation of a mixture of dyes and its use in a column for the same purpose were also investigated. Desorption was investigated using methanol and 90 min of reaction time.

An adsorbent of conjugated sponge of karaya gum and chitosan had been synthesized and used in the removal of anionic and cationic dyes from aqueous solutions [75]. Results from the investigation showed that the adsorbent presented an adsorption capacity of 33 mg/g for methyl orange and also 33 mg/g in the case of methylene blue. The aqueous pH value influenced the adsorption of methylene blue, decreasing it as the pH increased from 5 to 10. Methyl orange uptake responded well to the Langmuir isotherm, indicating adsorption on the homogeneous surface against methylene blue adsorption, which fitted to the Freundlich isotherm, indicative of an adsorption onto a heterogeneous surface. Both dyes in the concentration range of 10–50 mg/L followed the pseudo-second-order equation. Adsorption was caused by (i) electrostatic interaction between SO_3_^−^ groups of methyl orange and NH_3_^+^ groups in the adsorbent; in the case of methylene blue, (ii) the interactions were between N^+^ groups in the dye and COO^−^ groups present in the adsorbent. Desorption was carried out with 0.1 M of HCl (methylene blue) or 0.1 M of NaOH (methyl orange) and a slight but continuous decrease (six cycles) in the adsorption–desorption rates was observed.

In the race to find low-cost adsorbents, a non-toxic biosorbent possessing a high-charge density and thermal stability was prepared by using hexametaphosphate as an ionic cross-linker. The as-obtained chitosan microspheres were use in the removal of malachite green and anionic reactive red-195 from waters [76]. While the adsorption equilibrium was fixed at 120 min (malachite green) or 60 min (reactive red-195), the adsorption was best described in the 25–45 °C range by the pseudo-second-order model, in addition to the Freundlich and Temkin isotherms. In studying the influence of the temperature on dye adsorption, the exothermic and non-spontaneous (malachite green), and endothermic and spontaneous (reactive red-195) nature of the adsorption processes was observed. No desorption data were included in this work.

Synthetic and functionalized amorphous silica, silica nanosheets, and silica nanoparticles, modified by hexadecyl trimethyl ammonium bromide and hexadecyl trimethoxysilane, were used to remove dyes from solutions [77]. The investigated dyes were malachite green, crystal violet, and bromophenol blue. In the case of malachite green, the dye uptake increased with the pH value from 3 to 9, whereas for crystal violet and bromophenol blue, the adsorption was better in the 5–9 pH values range. Typical uptakes were 423 mg/g (malachite green), 409 mg/g (crystal violet), and 262 mg/g (bromophenol blue). Experimental data fitted to the pseudo-second-order and Freundlich models, attributable to the significant role of π–π stacking. No desorption data were included in this work.

In the next investigation, a microporous anionic metal–organic framework, JUC-210, was synthesized using a spirobifluorene-based ligand and In(III) [78]. Due to its structure, which consisted of a two-fold interpenetrated pts framework with a large void space, the material was investigated to separate various dyes. The adsorbent showed a high selective adsorption towards methylene blue and in the presence of neutral red, Sudan I, Orange II, and methyl orange. The adsorption of methylene blue followed the Lagmuir and pseudo-second-order kinetic models, and its adsorption, due to its positive charge, was based on ion-exchange and size exclusion. The desorption step was not considered in this work.

Methylene blue and methyl orange were adsorbed using a three-dimensional graphene aerogel material, fabricated from graphene oxide sheets with ethylenediamine acting as a reducing agent [79]. Maximum adsorption capacities were 222 mg/g (methylene blue) and 167 mg/g (methyl orange). The adsorption followed the Langmuir isotherm and pseudo-second-order kinetic model, with the maximum adsorption reached at pH 6 (methylene blue) or pH 3 (methyl orange). Adsorption equilibrium was achieved at very different times: 250 min in the case of methyl orange or near 800 min for methylene blue. No desorption data appeared in this work.

A mixed compound of nitrilotriacetic acid β-cyclodextrin-chitosan was produced for the simultaneous removal of dyes and metals [80]. In this adsorption process, β-cyclodextrin cavities adsorbed methylene blue by host/guest contacts and the remaining functional groups acted as adsorption sites for methyl orange (and metal ions). Maximum adsorption capacities of the adsorbent were 133 mg/g (methyl orange) and 163 mg/g (methylene blue). The adsorption data fitted to Sips (methyl orange) and Langmuir (methylene blue) models, and both dyes followed the pseudo-second-order kinetic model. Desorption was accomplished with HCl, ethanol, or nitric acid, followed by washing with deionized water. In binary solutions containing methylene blue and Hg(II), the removal efficiency for the dye was maintained nearly constant (95%) after four cycles.

Melamine and cyanuric chloride were directly used to fabricate a covalent triazine framework through a polycondensation; due to its properties, including numerous basic nitrogen atoms (near 59% wt %), the BET surface area (670 m/g), and the hierarchical pore structure, the material was investigated as a potential adsorbent of anionic dyes, using the removal of Congo red as an example [81]. The maximum adsorption capacity was of 1581 mg/g at 30 °C, with the adsorption mechanism attributable to the electrostatic attraction and hydrogen bonding between the dye and the adsorbent. As can be seen from Table 7, the adsorbent lost its effectiveness as the pH value increased.

The adsorption of cationic dyes can be improved when mixed with Congo red, e.g., single cationic dye solutions of malachite green or methylene blue resulted in an adsorption rate of 57% or 16%, respectively; however, the binary mixture of these with Congo red resulted in 100% (malachite green) or 92% (methylene blue) adsorption rates. This synergistic effect was attributed to (i) the preferential adsorption of Congo red onto the adsorbent surface, which reduced the surface charge and thus this surface was more adequate to adsorb cationic molecules, and (ii) to the fact that sulfonic groups in the Congo red molecule reacted with the positive groups (e.g., quaternary ammonium group) present in the cationic compound, enhancing their removal from the solution, i.e., Congo red acted as a bridge between the adsorbent and the cationic molecule. Congo red was desorbed with the use of a sodium chloride saturated ethanol solution for 4 h, for which the adsorption efficacy decreased from 100% to 90% after six cycles. Moreover, monolithic aerogels were produced by incorporating the adsorbent into polyvinylidene fluoride and through a further casting in melamine resin foams.

The next reference investigated the removal of methylene blue and orange green by a series of adsorbents polymers [82]. The adsorption of methylene blue reached the equilibrium after 45 min at pH 6.86 and 20 °C. This reference is unique because it included an approach to the cost of the adsorbents studied in the removal of dyes. Table 8 presents some of these results. No desorption data were included in this reference.

Jute fiber was subjected to a chemical activation method to yield activated carbon fibers [9]. The as-prepared material was used to investigate its adsorbent properties against methylene blue and 4-nitrophenol.

A foam, described as GO/g-C_3_N_4_/TiO_2_, was prepared via hydrothermal treatment and freeze-drying methods [83]. Besides oil–water mixtures, this material was used to adsorb organic dyes such as rhodamine B, methylene blue, and methyl orange. Both rhodamine B and methylene blue were adsorbed with efficiencies near 98%, whereas in a mixture of methylene blue and methyl orange, the cationic dye was totally adsorbed by the foam but methyl orange was maintained in the solution. No desorption data were included in this investigation.

Hydrothermal procedures were used to fabricate coordination polymeric chains modified by polyoxometalate, H_3_K_2_[Ag_5_(DTB)_5_][SiW_12_O_40_]_2_·Cl_2_·8H_2_O, where {DTB = 1,4-di(1H-1,2,4-triazol-1-yl)benzene} [84]. In single solutions, the removal rate for crystal violet was 74% after 90 min, and in the same time, 85% of methylene blue was removed from the solution. Using binary solutions, the material can separate methylene blue and crystal violet from methyl orange in a selective form; that is, separate cationic dyes from anionic ones. The data indicated that this adsorption was attributable to the sizes of the organic dyes. No desorption data were included in the manuscript.

A titanate-based material (peroxide sodium titanate, PST) was used as a precursor to fabricate three types of surface-charged surfactants: dodecyl dimethyl betaine (BS-PST), sodium dodecyl sulphate (SDS-PST), and dodecyltrimethyl ammonium chloride (DTAC-PST [85]. These were used to investigate the removal of contaminants such as methylene blue, acid red G, and ammonia nitrogen and phosphate. The batch experimentation showed that the DTAC-PST material had the best removal performance over methylene blue, though the maximum adsorption capacity was 82 mg/g (methylene blue) and 546 mg/g (acid red G). In simulated waters containing 50 mg/L of methylene blue and 50 mg/L of acid red G, in addition to the inorganic contaminants mentioned above, the dyes’ concentrations in the solution at the equilibrium were maintained below the limiting value of 10 mg/L, marked by the discharge levels in China (GB, 18918–2002) by using the adsorbent and controlling the pH value: 2–4 for acid red G or 2–10 for methylene blue. Electrostatic attraction and ligand exchange were the mechanisms of methylene blue uptake onto the adsorbent, whereas in the case of acid red G, C–N^+^ from the DTAC modification was responsible for the adsorption of the dye. The dye-loaded DTAC-PST could be regenerated by 0.5 M of NaOH solutions, maintaining near 80% of its adsorption capacity after five cycles.

Methylenen blue and rhodamine B were removed from solutions by the use of porous BN nano/microrods [86]. The material also acted in a selective form towards cationic dyes in the case of binary mixtures of Congo red and the above cationic dyes. This selectivity was attributable to the synergistic effect of (i) the electrostatic and π–π interactions; (ii) the size of the dye molecule; and (iii) the specific surface area and pore size of the adsorbent. The removal of the adsorbed dye from the adsorbent was not included in this investigation.

## 5. Conclusions

This review demonstrates that the subject regarding the removal of hazardous organic dyes from solutions is of considerable interest for numerous investigation groups and institutions around the world. The adsorbents used for this task contemplated the removal of solo anionic or cationic dyes, as well as the removal of mixtures of both types. These adsorbents presented different origins. Thus, one can find adsorbents based on metal composites and nanocomposites; magnetic or non-magnetic compounds; carbon-based compounds; metal–organic polymers; natural materials; silica-based compounds; organic-based compounds; MOF-based materials; graphene-based materials; and biomaterials, among others. In any case, the disparity, with respect to the dye uptake onto the adsorbent, is great. In this period of time, the removal of about 36 different organic dyes have been investigated; however, as can be seen from Table 9, the quantitative results showed by the papers greatly varied concerning the maximum dye adsorption capacity resulting from the experimental data.

However, as it can be seen in Table 10, this review demonstrates that the experimental investigations published during the first six months of the 2021 year were mainly polarized in a limiting geographical zone. Effectively, the vast amount of these investigations came from Asia and within this continent, authors and institutions came from China, exhibiting great differences over others.

Considering that many of the investigations claimed a process to remove dyes, it was noted that from the papers published, an important 39% of them did not present desorption results. In this context, the opinion of the reviewers is that this is less than acceptable because authors must realize that, as important as the adsorption of a given dye onto an adsorbent is, the release (desorption step) of the solute from the loaded adsorbent is equally important to recognize the real practical usefulness of the given adsorbent; that is, only with the adsorption step can one have 50% of the information necessary to judge the overall effectiveness of the adsorbent to remove a dye from solutions.

Though all the investigations were important to gain knowledge from an applied or even basic research point of view, in all the cases, the present reviewers detect that not one of the reported investigations mentioned what to do with the desorbed solution containing the desorbed dye, which presumably may have a greater dye concentration that the dye-bearing feed solutions, as normally the desorbed phase used lower volumes than the fed one. Thus, they become more concentrated in the dye and consequently are more hazardous or toxic than the initial ones.

Only one [20] of the reviewed manuscripts investigated the influence of the stirring speed on the dye uptake onto the adsorbent. This is an experimental variable often neglected by authors and may be of the same importance as others, e.g., pH, temperature, dosages, etc., because, as also mentioned in [20], with the correct stirring speed applied on the system, the thickness of the solution boundary layer reaches a minimum and the adsorption maximizes. Authors must note that the influence of this variable on dye adsorption (or lack of) is not predictable and only can be known after experimentation. Considering the above, must one consider correcting the quantitative results given in the manuscripts.

In any case, the topic concerning the removal of organic dyes from solutions by means of adsorption methodologies is still of fascinating interest to many research groups and by no doubt the number of adsorbents to be used in this field is expected to expand. Consequently, the results, hopefully considering the above points, will continue to be published and fill our knowledge-base in this important environmental and sociological issue.

## Figures and Tables

**Table 1 molecules-26-05440-t001:** Dye uptakes onto the various adsorbents tested.

Adsorbent	[MO]_experimental_, mg/g	[MO]_model_, mg/g
1Z:1N without extract	4.4	4.4
3Z:1N	5.4	5.9
1Z:1N	5.1	5.8
1Z:3N	12	11

Temperature: 20 °C; pH: 4; and time: 2 h. Abbreviation: MO, methyl orange.

**Table 2 molecules-26-05440-t002:** Efficacy of the adsorbent after cycles of adsorption–desorption.

Dye	1st Cycle	2nd Cycle	3rd Cycle	4th Cycle	5th Cycle
Reactive blue 4	85	83	70	65	60
Indigo carmine	90	85	80	73	65
Acid blue 158	93	85	80	65	63

**Table 3 molecules-26-05440-t003:** Effect of temperature on the theoretical maximum adsorption capacity.

Temperature, °C	[MB]_ad, m_, mg/g
20	228
25	233
30	237

Abbreviation: MB, methylene blue.

**Table 4 molecules-26-05440-t004:** Equilibrium capacities of the adsorbent.

Dye	[Dye]_ad,e_, mg/g	Dye	[Dye]_ad,e_, mg/g
Methylene blue	252	Crystal violet	1.6
Toluidine B	226	Rhodamine B	2
Methyl orange	47	Sodium fluorescein	4.2

pH 10. Time: 30 min.

**Table 5 molecules-26-05440-t005:** Performance of the adsorbent at various activations and methods of activation.

Adsorbent	Activation	Method	Days of Activation	% MB Adsorption	% MR Adsorption
AS-5	Acetone	Soxhlet	5	19	93
CS-5	Chloroform	Soxhlet	5	62	82
ES-5	Ethanol	Soxhlet	5	41	95
AC-5	Acetone	Centrifugation	5	43	90
CC-5	Chloroform	Centrifugation	5	80	94
EC-5	Ethanol	Centrifugation	5	44	85

Abbreviations: MB, methylene blue and MR, methyl red.

**Table 6 molecules-26-05440-t006:** Adsorption of dyes by ADES-1.

Dye	Removal Efficiency, %	[Dye]_ad.e_, mg/g
Methyl violet	86	1.8
Methylene blue	80	1.7
Rhodamine B	60	1.3
Methyl orange	54	0.99

Initial dye concentration: 5 × 10^−5^ M. Time: 3 h.

**Table 7 molecules-26-05440-t007:** Influence of the pH on the adsorption of anionic and cationic dyes.

Dye	Type	pH	[dye]_ad,m_, mg/g
Congo red	Anionic	5	1581
		12	600
Rhodamine B	Cationic	5	370
		12	250

Temperature: 30 °C. Time: 12 h. Dye concentrations: 60 mg/L. Adsorbent dosage: 30 mg/L.

**Table 8 molecules-26-05440-t008:** Approximate costs in the removal of methylene blue from solutions using various adsorbents.

Adsorbent	Adsorbent Cost, US Dollars	Methylene Blue Capacity, mol/g
Wheat flour	<5	3.3
Turmeric powder	10	6.8
Polyaniline	25–30	7.1
Nano(Fe-Ni-Si) oxide	near 10	22

Treatment of 1000 L of a synthetic solution containing 1 × 10^−5^ M dye.

**Table 9 molecules-26-05440-t009:** Some quantitative results concerning the organic dyes’ uptake onto different adsorbents.

Methylene Blue	Reference	Malachite Green	Reference	Methyl Orange	Reference	Congo Red	Referene
12	[40]	140	[38]	6	[61]	89	[73]
25	[32]	2000	[59]	133	[80]	140	[17]
573	[63]			255	[21]	331	[69]
1584	[43]			316	[25]	926	[29]
1690	[33]			1146	[26]	1250	[22]
1807	[44]			1409	[19]	1581	[81]

(Concentrations in mg/g).

**Table 10 molecules-26-05440-t010:** Distribution of the papers published in the first six months of the 2021 year.

Zone	Percentage ^a^
Africa	2.6
America	6.5
Asia	85.7
Europe	3.9
Oceania	1.3

^a^ Over a total of 77 manuscripts. Distribution based on the address and affiliation of the corresponding author.

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
