# Peer review of "Organic Dyes versus Adsorption Processing"

_molecules, 2021, doi:10.3390/molecules26185440_

Round 1

Reviewer 1 Report

The present review gives a summary of the adsorbents and adsorption techniques for organic dyes in the first half of 2021 year. The manuscript is well organized and the topic is attractive. I recommend that the manuscript can be accepted in the present form because the authors have made great improvements.

Author Response

Thank you for your comments.

Reviewer 2 Report

In this work authors present a review on the recent papers published about the removal of organic dyes from waste waters all over the world. Although the paper would be interesting I feel the weak information given about the recent methods or about the processes which would be the basis of the latest methods developed on this field. Statistics cannot be the focus of such papers and conclusion about the geographic distribution cannot based simply on the address of the corresponding authors. Due to this reasons I suggest major revision which mainly focus on the new, recently published scientific methodologies instead statistical evaluation.

Minor remarks:

lines 961-967: Desorption important, but mainly in cases when the adsorbent recyclable or it is necessary due to the further manipulation of the cleaning processes.

lines 968-973: Such statements do not give new information and the related processes are important in the cases of applied research. I suggest authors to discuss the papers according to their main message associated to the basic or applied research. 

Author Response

Please note that the present manuscript is a resubmission of a previous one taking into consideration the opinion of one of the reviewers (most probably reviewer 1 for the present manuscript), thus, we decided to classify the data on the basis of the type of the investigated dyes. However, and considering your comment, we added here a short summary of the type of adsorbents used in the investigations.

Please also noted, that the statistic, mentioned by yours, was done on the basis of the address and thus affiliation of the corresponding author: institution, city, country.   

Minor remarks:

lines 961-967: Desorption important, but mainly in cases when the adsorbent recyclable or it is necessary due to the further manipulation of the cleaning processes.

This is true, but please also note that many, if not most of the papers, claimed for adsorbents (processes) to remove dyes: without the desorption step the process was not complete. 

lines 968-973: Such statements do not give new information and the related processes are important in the cases of applied research. I suggest authors to discuss the papers according to their main message associated to the basic or applied research.

Based also in the above: a process,  authors did not realize that the problem was not only to remove the hazardous dye from waters, but also what to do with the desorbed solution. Since at the end a solution more hazardous that the initial feed was yielded, as it contained a greater dye concentration of the initial one. If one not considered what to do, only the dye was passed from one solution to another, and the problem of the toxicity was not resolved.  

Round 2

Reviewer 2 Report

Maybe it is a subjective opinion, but inserting the paragraph into the Conclusion have improved lot the manuscript, especially to understand the main message. So now I am satisfied with the manuscript in the present form and suggest publication as it is. 

This manuscript is a resubmission of an earlier submission. The following is a list of the peer review reports and author responses from that submission.

Round 1

Reviewer 1 Report

The review article summarizes the advances in adsorption materials and techniques for organic dyes in the past half-year period. The topic of the review is interesting because the removal of organic dyes from water is still attracting scientists’ attention worldwide, although organic dyes are sometimes employed as model molecules instead of target molecules in some research work. Therefore, I recommend the paper can be accepted after major revision.

  1. The title “Results”(Line 37) is inappropriate in a review paper. Please revise it.
  2. The main body of the article is focusing. I suggest that this section can be re-organized by adsorbents, dyes, or adsorption mechanisms.
  3. As a review article, some summative graphics of high quality should be included. However, the paper has no any figures.
  4. Tables 1-8 are extracted from original papers, and a summative table should be supplemented to compare the capacity and efficiency of the adsorbents.
  5. I recommend that the specific surface area, pore volume, pore diameter of adsorbents should be studied systematically to analyze the adsorption capacity of the dyes.
  6. The nature of the adsorbent itself will also have a certain impact on the amount of adsorption. However, this influence factor does not give an exact analysis and comparison in the manuscript.
  7. I suggest authors should add all abbreviations and their full names should appear in your manuscript.
  8. At present, the manuscript is mainly focused on the first half of 2021, and the amount of research is relatively small enough to be conclusive.
  9. Please correct any minor errors in molecular formula Fe3O4in Line 56, and Zr_MOC-2in Line 85, the ([Cu(bmpae)0.5(Mo3O10)(H2O)]·H2O) in Line 92, the (Fe3O(OH2)3Cl(NH2-BDC)3·9.5H2O) in Line 223, temperature unit (Line 655 and Line 662), punctuation, etc. in the contents in your manuscript.
  10. Grammar and some chemical valence representations need to check throughout the manuscript. There are many small mistakes and inconsistencies in the manuscript, so the authors are requested to read and correct them carefully.

Author Response

The review article summarizes the advances in adsorption materials and techniques for organic dyes in the past half-year period. The topic of the review is interesting because the removal of organic dyes from water is still attracting scientists’ attention worldwide, although organic dyes are sometimes employed as model molecules instead of target molecules in some research work. Therefore, I recommend the paper can be accepted after major revision.

Thank you for you comments.

The title “Results”(Line 37) is inappropriate in a review paper. Please revise it.

We changed it.

The main body of the article is focusing. I suggest that this section can be re-organized by adsorbents, dyes, or adsorption mechanisms.

We try to do as you mentioned, but that since that i) there are many dyes, ii) there are many adsorbents and probably the most determinant iii) in many papers, the researchers used cationic, neutral and anionic dyes, thus, making a classification seemed to do very difficult, and we opted by wrote the article as it was done.    

As a review article, some summative graphics of high quality should be included. However, the paper has no any figures.

That is true, we liked much to summarize some results as Tables than Figures.

Tables 1-8 are extracted from original papers, and a summative table should be supplemented to compare the capacity and efficiency of the adsorbents.

Tables 1-8 were written from results appearing in the respective manuscript, not directly from the manuscript. As we mentioned above, the adsorbents were of very different nature, thus, a comparison was not, in our opinion, possible, since the experimental conditions, and how the results were obtained, was far to be the same. In Table 10 we compare some experimental results derived from the removal of some dyes, and you aware the overall variety of concentrations reached on the respective adsorbent.

I recommend that the specific surface area, pore volume, pore diameter of adsorbents should be studied systematically to analyze the adsorption capacity of the dyes.

This was other problem that we find when consider what to present in the review, since practically the properties that you mentioned above were specific for each manuscript. Thus, we prefer that in the case that a potential reader will be interested for any given reference, it will go to the original manuscript to find completel information about the particular system.  

The nature of the adsorbent itself will also have a certain impact on the amount of adsorption. However, this influence factor does not give an exact analysis and comparison in the manuscript.

Sure it have, however for the reasons mentioned above we did not mix information (comparison) from one adsorbent to another, except in Table 10, and you see the variety of results on the same dyes.

I suggest authors should add all abbreviations and their full names should appear in your manuscript.

In the revised manuscript we tried to do the above.

At present, the manuscript is mainly focused on the first half of 2021, and the amount of research is relatively small enough to be conclusive.

Since the manuscript must reach the Editorial Office in June 2021, we decided not to present published data more beyond this date.

Please correct any minor errors in molecular formula Fe3O4in Line 56, and Zr_MOC-2in Line 85, the ([Cu(bmpae)0.5(Mo3O10)(H2O)]·H2O) in Line 92, the (Fe3O(OH2)3Cl(NH2-BDC)3·9.5H2O) in Line 223, temperature unit (Line 655 and Line 662), punctuation, etc. in the contents in your manuscript.

We amended the above.

Grammar and some chemical valence representations need to check throughout the manuscript. There are many small mistakes and inconsistencies in the manuscript, so the authors are requested to read and correct them carefully.

We amended the above.

Reviewer 2 Report

The manuscript seems to be a well-written and comprehensive review on the related field. I recommend for publication.

Author Response

This paper presents a comprehensive review of the very latest adsorption studies published in the first half of 2021. Because of the large number of published papers on adsorption research, researchers in the relevant field often find it difficult to gather all the information. This kind of review is worthy of periodic publication because it helps researchers in the relevant field stay up-to-date on the latest information. One thing I would like the authors to consider is that although desorption of pollutants is important from a practical point of view, it is still important to quantitatively study and report how much new materials can adsorb from a scientific point of view. In that respect, I think it's unnecessarily negative to say "being not acceptable" for not presenting desorption results in many papers.

Thank you for your kind and positive comments about our manuscript. Please let that we have a little discrepancy to your comment about adsorption-desorption steps and the lack of information about the latter in some manuscripts. It is right that available information about adsorption is always welcomed, however, and this is our point of view after near 40 years of hydrometallurgical experience, when authors wrote about “processing” and/or “removal” of contaminants, one must have in mind that as important to achieve the removal of the toxics from the feed solution, it is to recover them from the adsorbent (also ion-exchange resins, liquid-liquid extractants) to complete the overall operation. Because if the above is not considered, the adsorbent, etc., can not be reused and probably the process will not be economically sound; also, as you aware, the adsorbent works well but may be degraded (i.e. dissolved) in the desorption step. Moreover, we detected that in most of the cases, the investigations do not mentioned nothing about what to do with the contaminant-bearing desorbed solution, and this may be another problem, since this last solution may contained a greater concentration of the contaminant (thus, becomes more hazardous that the feed phase). We hope that you agreed, at least partially, with that written above.

In any case, we amended the comment in relation with yours one.             

Round 2

Reviewer 1 Report

I reviewed the revised manuscript and found that the authors made a bit changes, but I still think that the revised manuscript needs more improvement. Therefore, I suggest major revision or resubmission.

(1) Authors changed the subtitle from “Results” to “Removal of dies from waters using adsorption processing: the publications”. What does “dies” mean here? The authors did not revise the manuscript carefully.

(2) I suggest that the main section should be re-organized by adsorbents, dyes, or adsorption mechanisms, which is very important for readers. But I cannot see any improvement. The authors tried to find some excuses to avoid doing this.

(3) Some summative graphics of high quality should be included in a review paper.